# Assessing swallowing disorders in adults on high-flow nasal cannula in critical and non-critical care settings. A scoping review protocol

**Ruvistay Gutierrez-Arias**[1,2]*, **Gabriel Salgado-Maldonado**[1,3,4], **Paola Letelier Valdivia**[1], **Francisco Salinas-Barahona**[1,5], **Carmen Echeverría-Valdebenito**[1], **Pamela Seron**[6,7], on behalf of INTRehab Research Group¶

**1** Departamento de Apoyo en Rehabilitación Cardiopulmonar Integral, Instituto Nacional del Tórax, Santiago, Chile, **2** Exercise and Rehabilitation Sciences Institute, Faculty of Rehabilitation Sciences, Universidad Andres Bello, Santiago, Chile, **3** Laboratorio de Neurociencias Cognitivas (LANNEC), Clínica de Memoria y Neuropsiquiatría (CMYN), Universidad de Chile, Santiago, Chile, **4** Millennium Nucleus to Improve the Mental Health of Adolescents and Youths, Imhay, Santiago, Chile, **5** Escuela de Kinesiología, Facultad de Ciencia de la Salud, Universidad Autonoma de Chile, Santiago, Chile, **6** Centro de Excelencia CIGES, Universidad de La Frontera, Temuco, Chile, **7** Departamento de Ciencias de la Rehabilitación, Facultad de Medicina, Universidad de La Frontera, Temuco, Chile

¶ Membership of the INTRehab Research Group is provided in the Acknowledgments.
* rgutierrez@tirax.cl

**Data Availability Statement:** No datasets were generated or analysed during the current study. All

## Abstract

### Introduction

The high-flow nasal cannula (HFNC) has become a widely used respiratory support system, which has proven to be effective in different populations. The facilitation of oral communication and feeding have been described as advantages of this support. Nevertheless, swallowing disorders associated with the use of HFNC have been postulated. However, such evidence is scattered in the literature, not systematically searched, and needs to be adequately summarised. This review aimed to explore the literature, to identify and map the evidence, regarding the frequency and methods of assessment of swallowing disorders in adult HFNC users, in both critical and non-critical units.

### Materials and methods

A scoping review will be conducted. A systematic search in MEDLINE (Ovid), Embase (Ovid), CENTRAL, CINAHL (EBSCOhost), and other resources will be conducted. Primary studies, in any language or publication status, assessing the incidence of swallowing disorders in adults with HFNC support will be included. Two reviewers will independently select studies and extract data. Disagreements will be resolved by consensus or a third reviewer. The results will be reported narratively, using tables and figures to support them.

relevant data from this study will be made available upon study completion.

**Funding:** The authors received no specific funding for this work.

**Competing interests:** The authors have declared that no competing interests exist.

## Discussion

Positive end-expiratory pressure generated in the airway by HFNC could impair the proper swallowing performance. Knowing the methodological characteristics, the instruments or scales used to assess the presence of dysphagia, and the results of the studies may contribute to considering swallowing assessment in this population on a routine basis, as well as to guide the conduct of new studies that may respond to less researched areas in this topic.

## Registration

Registration number: INPLASY2022110078.

## Introduction

The high-flow nasal cannula (HFNC) has become a widely used respiratory support system [1], increasing in use due to the COVID-19 pandemic [2]. A number of studies have reported the effectiveness of HFNC, especially in hypoxemic respiratory failure, to avoid invasive mechanical ventilation [3–5] and the weaning process to avoid respiratory failure post-extubation [3, 6, 7]. The main mechanisms that explain its effects are airway humidification, decreased resistance, flushing of the upper airway dead space, ability to deliver $O_2$ at planned concentrations, and the possibility of generating positive end-expiratory pressure [8].

Several advantages of using HFNC compared to conventional oxygen masks [9–11] and non-invasive mechanical ventilation have been described [12–14]. Increased comfort is frequently mentioned by people undergoing HFNC, as adequate communication and oral feeding are allowed during its use [15–17].

Despite the advantages of using HFNC, some minor adverse effects have been reported, the most common being epistaxis and nasal discomfort associated with dryness [18]. However, the possibility of more serious adverse events, such as aspiration pneumonia associated with using HFNC, has been postulated [19].

Most of the research related to the research question posed in this protocol has focused on pediatrics, probably because the evidence supports the use of HFNC in this population [20, 21]. Evidence synthesis studies that have considered this topic in the adult population have not focused on methods of assessment disorders and have not been conducted using a systematic methodology [22, 23]. In addition, have limited the inclusion of studies by publication status [22], and have presented findings by summarizing studies separately rather than in an integrative analysis [22].

There are currently different primary studies using diverse assessment methodologies, which report disparate results on the incidence of swallowing disorders. However, such evidence is scattered in the literature, not systematically searched, and needs to be adequately summarized. Therefore, the findings of this scoping review would provide insight into the methodological characteristics, the frequency of swallowing disorders, and the instruments or scales used to assess their presence. This could be used as a basis for further analytical studies, or systematic diagnostic reviews, that could address this less researched area, and ultimately contribute to considering swallowing assessment in this population on a routine basis.

This review aimed to explore the literature to identify and map the evidence regarding the frequency and methods of assessing swallowing disorders in adult HFNC users in critical and non-critical units.

## Materials and methods

A scoping review will be conducted following the updated recommendations of the Joanna Briggs Institute (JBI) [24]. The protocol for this review was registered on the International Platform of Registered Systematic Review and Meta-analysis Protocols (INPLASY) under the number INPLASY2022110078, and it was reported following Preferred Reporting Items for Systematic Review and Meta-analysis Protocols (PRISMA-P) (S1 Checklist) [25]. The results will be reported following the Extension for Scoping Reviews of the Preferred Reporting Items for Systematic Reviews and Meta-analyses statement (PRISMA-ScR) [26].

### Search strategy

A systematic search will be conducted in MEDLINE, through the Ovid platform; Embase, through the Ovid platform; Cochrane Collaboration Central Register of Controlled Trials (CENTRAL), through the Cochrane Library; and Cumulative Index of Nursing and Allied Literature(CINAHL), through the EBSCOhost platform. The strategy will consider a sensitive approach, controlled language (MeSH, EMTREE, CINAHL Subject Heading), and natural language. The strategy to be used for MEDLINE-Ovid will be adapted to construct the search in the other databases (Table 1).

In addition, clinical trial registries will be searched (https://clinicaltrials.gov/ and https://trialsearch.who.int/), and references from reviews related to the objective of this scoping review and included studies will be evaluated.

### Eligibility criteria

Eligibility criteria for study selection will be divided into participants or populations included in the studies, the concept or phenomenon involved, and the context in which the studies were

**Table 1. Search strategy for MEDLINE using the Ovid platform.**

| N˚ | Search term |
| --- | --- |
| 1 | exp Oxygen/ |
| 2 | exp Oxygen Inhalation Therapy/ |
| 3 | Cannula/ |
| 4 | (1 or 2) and 3 |
| 5 | ((high flow or highflow or high-flow or high frequency or nasal$) adj6 can?ul$).af. |
| 6 | ((high flow or highflow or high-flow or high frequency or prong$) adj6 nasal$).af. |
| 7 | ((high flow or highflow or high-flow or high frequency) adj4 oxygen$).af. |
| 8 | (HFNC or HFNP or Vapotherm or Optiflow, or Respircare).af. |
| 9 | transnasal insufflation.af. |
| 10 | or/4-9 |
| 11 | Deglutition/ |
| 12 | exp Deglutition Disorders/ |
| 13 | (swallow$ or deglutit$ or dysphag$).af. |
| 14 | exp Respiratory Aspiration/ or exp Pneumonia, Aspiration/ |
| 15 | (inhal$ or aspirat$ or ingest$).af. |
| 16 | Pharynx/ or pharyngeal muscles/ or esophageal sphincter, upper/ or exp Esophagus/ |
| 17 | (throat or oesophag$ or esophag$ or pharyn$ or oropharyn$).af. |
| 18 | or/11-17 |
| 19 | 10 and 18 |

conducted (PCC framework) [24]. In addition, the design of the studies will be considered for inclusion in this review.

**Participants.** We will include studies that have recruited adults (18 years or older). These may be healthy people who have voluntarily participated in studies to evaluate the use of HFNC or have a disease requiring acute or chronic use of HFNC.

**Concept.** Studies evaluating swallowing disorders during the application of HFNC will be included. The tools or scales used to assess dysphagia, the programmed flows in HFNC, and the resulting inspired $O_2$ fraction will not limit the inclusion of studies. Any HFNC device or equipment will be considered eligible.

**Context.** Studies conducted in the hospital, post-discharge care, or home care settings will be included. Studies conducted in research centers will also be considered. If HFNC is used as a method to avoid invasive mechanical ventilation or in the context of preventing post-extubation failure, it will not limit the inclusion of studies.

**Study designs.** Primary studies (clinical trials, cohort studies, case-control, cross-sectional, and case reports) will be included. Regarding publication status, studies reported as full text or abstracts presented in conference proceedings will be included. The language, as well as the publication date of the studies, will not limit their inclusion. For the translation of studies published in languages other than English and Spanish, researchers who are native speakers of the language of publication will be contacted to resolve doubts regarding the general context of the study.

## Selection of studies

Once the studies have been searched, duplicates will be removed using the Mendeley® reference manager (Mendeley Desktop Version 1.19.8 - Elsevier Inc.) and the Rayyan® app [27]. Titles and abstracts will be independently screened by two research team members, who will discard studies irrelevant to this review. Subsequently, the full texts of the potential investigations to be included will be analyzed to determine which articles meet all the eligibility criteria. The Rayyan® app will be used for this stage [27].

In the first instance, disagreements will be resolved by consensus, and if they persist, a third reviewer will determine the inclusion of the studies.

## Information extraction

Two reviewers will independently extract information from the included studies. An extraction form specifically designed to meet the objectives of this review will be used and developed in a Microsoft Excel® spreadsheet.

The information to be extracted will include aspects related to the characteristics of the publications and studies, as well as the population, HFNC devices or equipment, HFNC therapy programming, frequency (prevalence or incidence) of swallowing disorder and their assessment instruments or scales, and authors' findings or conclusions (Table 2).

Data extraction will be carried out by 4 researchers in pairs of one experienced and one novice reviewer. In the first instance, disagreements will be resolved by consensus, and if they persist, a third reviewer will determine the inclusion of the studies.

## Synthesis of information

The search results and selection of studies will be reported through a PRISMA flow chart [28]. In addition, the reasons for excluding full-text evaluated studies will be reported in a table. Findings will be reported for all studies and separated by those conducted in critical and non-critical settings. The critical setting will comprise the intensive critical units, and the non-critical setting will comprise all other units and the out-of-hospital.

**Table 2. Description of data to be extracted.**

| Information | Description |
|---|---|
| Identification of the studies | Title of the study, journal name, year of publication, authors' names, and authors' nationality. |
| Population | Age, condition that led to the use of HFNC, and time of use of HFNC |
| HFNC | Device and parameters (flow, temperature level, FiO$_2$) |
| Dysphagia assessment | Professional conducting the assessment, assessment protocol, assessment tool, or scale. |
| Author's findings | Incidence and severity of dysphagia according to flow used and authors' conclusions. |

**HFNC:** High-flow nasal cannula; **FiO$_2$**: Fraction inspired of O$_2$

The results will be reported in narrative form, and tables and figures will synthesize the information. Waffle charts [29], or similar, will represent the different primary study designs included.

## Ethics and dissemination

As a synthesis of evidence, this study does not involve the participation of people whose rights may be violated. However, this scoping review will be developed rigorously and systematically to achieve valid and reliable results.

The findings of scoping review will be presented at conferences and published in a peer-reviewed journal related to critical care or speech therapy.

## Discussion

Positive end-expiratory pressure generated in the airway by HFNC, with varying values depending on the flow used, could impair the proper swallowing performance, altering synchronization of breathing and swallowing [22]. This could be explained by the fact that during normal swallowing, the contact time phase of the vellum against the posterior pharyngeal wall is delayed. It is possible that positive pressure increases this delay and causes aspiration through the vellum and opening of the rear pharyngeal wall [19].

The results of this scoping review will provide insight into published studies to determine a possible relationship between the use of HFNC and the presence of swallowing disorders. Knowing the methodological characteristics, the relative dysphagia frequency and the instruments or scales used to assess the presence of dysphagia and the results of the studies may guide the conduct of new studies that may respond to less researched areas in this topic and contribute to considering swallowing assessment in this population on a routine basis.

## Supporting information

**S1 Checklist. PRISMA-P 2015 checklist.**
(DOCX)

## Acknowledgments

The members of the INTRehab Research Group are Adriana Lastra Morales, Mauricio Contador Pastene, Marjorie Valdés Araneda, Osvaldo Cabrera Román, Cristián Olave Contreras, Paula Herrera Torres, Francisco Salinas Barahona, Rocío Salazar Invernizzi, Ruvistay Gutiérrez Arias, Valeria Rivas Gálvez, Macarena Cerda Magna, Roberto Vergara Cabezas, Geraldine Castañeda Ayala, Paulina Wilson Meyer, Teresita Cortés Trivelli, Martina Angulo Henríquez,

Carmen Echeverría Valdebenito, Mathias Olivares Álvarez, Sebastián Calderón Fuentes, Victoria González Berrios, Iván Ramírez Venegas, Ariela Moreno Huircaleo, Ignacio Cortéz Barriento, Karim Alul Araya, Melanin Aldunce Saavedra, Sofía Keppeler Bertolotto, Natalia Gahona Estay, Andrea Cárdenas Castaño, Sthephanie Rodriguez Norambuena, Natalia Guajardo Latorre, Olenka Villlalón Banderas, Angel Castro Becerra, Diego Ibacache Huerta, Pablo Gómez Soto, María Paz Riquelme Velásquez, Constanza Toro García, Sthefany Quezada Hernández, Gabriel Salgado Maldonado, Paola Letelier Valdivia, Catalina Orellana Molnar & Katherine Peralta Arancibia.

## Author Contributions

**Conceptualization:** Ruvistay Gutierrez-Arias, Gabriel Salgado-Maldonado, Paola Letelier Valdivia, Francisco Salinas-Barahona, Carmen Echeverría-Valdebenito, Pamela Seron.

**Methodology:** Ruvistay Gutierrez-Arias, Pamela Seron.

**Project administration:** Ruvistay Gutierrez-Arias.

**Supervision:** Ruvistay Gutierrez-Arias.

**Writing – original draft:** Ruvistay Gutierrez-Arias.

**Writing – review & editing:** Gabriel Salgado-Maldonado, Paola Letelier Valdivia, Francisco Salinas-Barahona, Carmen Echeverría-Valdebenito, Pamela Seron.

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
