## [Decision Letter · Decision Letter 0]

29 Jun 2023

PONE-D-23-04634Assessing swallowing disorders in adults on high-flow nasal cannula in critical and non-critical care settings. A protocol for a scoping reviewPLOS ONE

Dear Dr. Gutierrez-Arias,

Thank you for submitting your manuscript to PLOS ONE. After careful consideration, we feel that it has merit but does not fully meet PLOS ONE’s publication criteria as it currently stands. Therefore, we invite you to submit a revised version of the manuscript that addresses the points raised during the review process.

ACADEMIC EDITOR: Please revise the manuscript according to the reviewers comments.

We look forward to receiving your revised manuscript.

Kind regards,

Silvia Fiorelli

Academic Editor

PLOS ONE

Journal Requirements:

The funders did not and will not have a role in study design, data collection and analysis, decision to publish, or preparation of the manuscript. 

Reviewers' comments:

Reviewer's Responses to Questions

**Comments to the Author**

1. Does the manuscript provide a valid rationale for the proposed study, with clearly identified and justified research questions?

Reviewer #1: Partly

Reviewer #2: Partly

2. Is the protocol technically sound and planned in a manner that will lead to a meaningful outcome and allow testing the stated hypotheses?

Reviewer #1: Partly

Reviewer #2: Partly

3. Is the methodology feasible and described in sufficient detail to allow the work to be replicable?

Reviewer #1: Yes

Reviewer #2: Yes

4. Have the authors described where all data underlying the findings will be made available when the study is complete?

Reviewer #1: Yes

Reviewer #2: No

5. Is the manuscript presented in an intelligible fashion and written in standard English?

Reviewer #1: No

Reviewer #2: Yes

6. Review Comments to the Author

You may also provide optional suggestions and comments to authors that they might find helpful in planning their study.

Reviewer #1: 1. Please descibe in full the methodology. Will any reference manager be used? What is the time frame? How is the distinction between critical and non-critical setting done? Are there two or three researchers engaged in the analysis of publications?

2. Figure 1 is not mentioned int the text, it also does not add any value to the publication.

3. The protocol requires english editing.

4. Line 32 and line 33 - two consecutive sentences start with "However". Please replace it with a different vocabulary.

5. Line 40 - the authors declare the research in any language will be considered. How are the authors planning to perform translation to understant the full context of the publication?

6. Line 44, line 166 - the term "hurt" seems to be ambiguous, please consider different term.

7. Line 45-48 - this sentence is too long and unclear, please consider rephrasing.

9. Line 56 - please consider adding a subject, e.g "especially in patients with hypoxemic respiratory failure" instead of effectivenes in [...], especially in hypoxemic respiratory failure

9. Line 59 - please consider using a different term instead of "known", "set" or "planned" seems to be more adequate verb here

10. Line 63 0 please consider adding "its" or "HFNC" before the word "use"

11. Line 70 - none of the cited studies [19-21] concerns newborns. In study 19 & 20 pediatric population is being studied, however very few newborns are included. Study 21 reffers mainly to adults.

12. Line 76 - please refrain from repeating an adjective "different" in one sentence.

13. Line 77-78 - the sentence "The flow used in HFNC needed to trigger them in adults with and without diseases in incomprehensible

14. Line 98 - 99 - repeated phrase "through the Ovid platform"

15. Line 116-118 phrase "adults [...] who may be healthy or have a disease requiring acute or chronic use of HFNC" is unclear. Are patents requiring HFNC without any underlying diseases or with acute or chronic resp. diseases included? Please clarify

16. Line 166-167 - "altering the preathing and swallowing pattern" - is altering the synchronization of breathing and swallowing ment?

17. Line 168-170 - the sentence "This could be explained [...]." is unclear, the term "vitellus" stands for an egg yolk, what was ment?

Reviewer #2: The authors present the protocol for a narrative review on swallowing disorders in adult patients with ongoing high flow nasal therapy. Authors properly state pitfalls of previous similar studies: not systematic searches or not adequately summarized.

Authors aim to review the frequency of swallowing disorders and scales used for assessment. Some confoundings are commented: flow, oxygen fraction, device, location of patients (critical vs non-critical), age.

My main concern is about management of these confoundings to present the results. Could authors detail?

Authors state that aspiration pneumonia has been associated with HFNC, but a cite is lacking.

7. PLOS authors have the option to publish the peer review history of their article (what does this mean?). If published, this will include your full peer review and any attached files.

Reviewer #1: **Yes: **Joanna Maria Jassem-Bobowicz

Reviewer #2: No

---

## [Author Response · Author response to Decision Letter 0]

11 Jul 2023

Journal Requirements:

Comment: 1. Please ensure that your manuscript meets PLOS ONE's style requirements, including those for file naming…

Response: The manuscript was edited according to PLOS ONE style requirements.

Comment: At this time, please address the following queries:

Response: In the “Rebuttal Letter” we requested the change of the funding statement to “The authors received no specific funding for this work.”

Comment: 3. Your ethics statement should only appear in the Methods section of your manuscript. If your ethics statement is written in any section besides the Methods, please move it to the Methods section and delete it from any other section. Please ensure that your ethics statement is included in your manuscript, as the ethics statement entered into the online submission form will not be published alongside your manuscript.

Response: The ethics statement was moved to the Methods section and removed from other parts of the manuscript.

Comment: 4. Please include captions for your Supporting Information files at the end of your manuscript, and update any in-text citations to match accordingly. Please see our Supporting Information guidelines for more information: http://journals.plos.org/plosone/s/supporting-information.

Response: Our Supporting Information subheading was included at the end of the manuscript and cited in the Methods section.

Reviewer #1

Comment: Please descibe in full the methodology. Will any reference manager be used? What is the time frame? How is the distinction between critical and non-critical setting done? Are there two or three researchers engaged in the analysis of publications?

Response: Thank you for your questions. Several methodological clarifications have been added to the manuscript:

* “Once the studies had been searched, duplicates will be removed using the Mendeley® (Mendeley Desktop Version 1.19.8 - Elsevier Inc.) reference manager and the Rayyan® app”.

* “Data extraction will be carried out by 4 researchers in pairs of one experienced and one novice reviewer”.

* “Findings will be reported for all studies and separated by those conducted in critical and non-critical settings. The critical setting will comprise the intensive critical units, and the non-critical setting will comprise all other units and the out-of-hospital”.

Comment: Figure 1 is not mentioned int the text, it also does not add any value to the publication.

Response: Thank you for your comment. We have replaced the figure with a table to detail the data that will be extracted from the included studies.

Comment: The protocol requires english editing.

Response: The protocol was reviewed and edited by a native English speaker.

Comment: Line 32 and line 33 - two consecutive sentences start with "However". Please replace it with a different vocabulary.

Response: Thank you for your comment. “However” has been changed to “Nevertheless”.

Comment: Line 40 - the authors declare the research in any language will be considered. How are the authors planning to perform translation to understant the full context of the publication?

Response: Thank you for your comment. In the Methods section, the following idea was added:

“For the translation of studies published in languages other than English and Spanish, researchers who are native speakers of the language of publication will be contacted to resolve doubts regarding the general context of the study.”

Comment: Line 44, line 166 - the term "hurt" seems to be ambiguous, please consider different term.

Response: Thank you for your comment.

Comment: Line 45-48 - this sentence is too long and unclear, please consider rephrasing.

Response: The term "hurt" was changed to "impair" in both sections of the manuscript.

Comment: Line 56 - please consider adding a subject, e.g "especially in patients with hypoxemic respiratory failure" instead of effectivenes in [...], especially in hypoxemic respiratory failure.

Response: Thanks for the suggestion. The sentence was replaced by “A number of studies have reported the effectiveness of HFNC”.

Comment: Line 59 - please consider using a different term instead of "known", "set" or "planned" seems to be more adequate verb here

Response: Thanks for the suggestion. “Known” has been changed to “planned”.

Comment: Line 63 0 please consider adding "its" or "HFNC" before the word "use"

Response: Thanks for the suggestion. Added the term “its”.

Comment: Line 70 - none of the cited studies [19-21] concerns newborns. In study 19 & 20 pediatric population is being studied, however very few newborns are included. Study 21 reffers mainly to adults.

Response: Thank you for your comment. Reference 21 was deleted, and the sentence was left as follows: 

“Most of the research related to the research question posed in this protocol has focused on pediatrics, probably because the evidence supports the use of HFNC in this publication.”

Comment: Line 76 - please refrain from repeating an adjective "different" in one sentence.

Response: Thank you for your comment. “different” has been changed to “diverse”.

Comment: Line 77-78 - the sentence "The flow used in HFNC needed to trigger them in adults with and without diseases in incomprehensible

Response: The phrase was deleted.

Comment: Line 98 - 99 - repeated phrase "through the Ovid platform"

Response: According to the PRISMA-S and PRESS declaration, the search platform is required for each database. In this case, we report that we will use OVID to search MEDLINE and Embase, as well as other platforms for the remaining databases.

Comment: Line 116-118 phrase "adults [...] who may be healthy or have a disease requiring acute or chronic use of HFNC" is unclear. Are patents requiring HFNC without any underlying diseases or with acute or chronic resp. diseases included? Please clarify

Response: Thank you for your comment. Let us explain the next sentence better: 

“These may be healthy people who have voluntarily participated in studies to evaluate the use of HFNC or have a disease requiring acute or chronic use of HFNC.”

Comment: Line 166-167 - "altering the preathing and swallowing pattern" - is altering the synchronization of breathing and swallowing ment?

Response: Thanks for the suggestion. “Altering the breathing and swallowing pattern” was changed by “synchronization of breathing and swallowing.”

Comment: Line 168-170 - the sentence "This could be explained [...]." is unclear, the term "vitellus" stands for an egg yolk, what was ment?

Response: We change the term "vitellus" to “vellum”.

Reviewer #1

Comment: Some confoundings are commented: flow, oxygen fraction, device, location of patients (critical vs non-critical), age. My main concern is about management of these confoundings to present the results. Could authors detail?

Response: The purpose of a scoping review is to describe what the literature says or what research has been done on a specific topic. Our aim is not to provide recommendations for clinical practice, which is rather the aim of systematic reviews and clinical practice guidelines, but to identify what has been studied regarding the possible association between the use of HFNC and swallowing disorders. Therefore, our study will serve to identify gaps in knowledge and provide recommendations for future research.

All possible variables that could confound the relationship between swallowing disorder and HFNC use will be discussed for future primary studies to consider.

Comment: Authors state that aspiration pneumonia has been associated with HFNC, but a cite is lacking.

Response: A reference was added to support the possibility that the use of HFNC may generate aspiration.

---

## [Decision Letter · Decision Letter 1]

6 Sep 2023

Assessing swallowing disorders in adults on high-flow nasal cannula in critical and non-critical care settings. A scoping review protocol

PONE-D-23-04634R1

Dear Dr. Ruvistay Gutierrez-Arias,

We’re pleased to inform you that your manuscript has been judged scientifically suitable for publication and will be formally accepted for publication once it meets all outstanding technical requirements.

Kind regards,

Silvia Fiorelli

Academic Editor

PLOS ONE

Additional Editor Comments (optional):

Congratulations to the authors and thanks to the reviewers for the suggestions provided which really helped improve the quality of the manuscript

Reviewers' comments:

Reviewer's Responses to Questions

**Comments to the Author**

1. Does the manuscript provide a valid rationale for the proposed study, with clearly identified and justified research questions?

Reviewer #2: Yes

Reviewer #3: Yes

2. Is the protocol technically sound and planned in a manner that will lead to a meaningful outcome and allow testing the stated hypotheses?

Reviewer #2: Yes

Reviewer #3: Yes

3. Is the methodology feasible and described in sufficient detail to allow the work to be replicable?

Reviewer #2: Yes

Reviewer #3: Yes

4. Have the authors described where all data underlying the findings will be made available when the study is complete?

Reviewer #2: Yes

Reviewer #3: Yes

5. Is the manuscript presented in an intelligible fashion and written in standard English?

Reviewer #2: Yes

Reviewer #3: Yes

6. Review Comments to the Author

You may also provide optional suggestions and comments to authors that they might find helpful in planning their study.

Reviewer #2: All my comments were adequately answered. The protocol has been clearly improved. I have no further comments.

Reviewer #3: Many thanks for asking me to review the above Scoping Reveiw.

The authors aimed to explore the literature to identify and map the evidence regarding the frequency and methods of assessing swallowing disorders in adult HFNC users in critical and non critical units.

It is a well written manuscript and the findings, provide insight into the methodological characteristics, the frequency of swallowing disorders, and the instruments used to assess their presence.

The authors have clearly and precisely responded to the comments of previous Reviewers, thus clarifying some "grey" areas of the Study's methodology.

Not having to point out any additional comment or correction, I think the Manuscript should be published in its current form.

7. PLOS authors have the option to publish the peer review history of their article (what does this mean?). If published, this will include your full peer review and any attached files.

Reviewer #2: No

Reviewer #3: No

---

## [Editor Report · Acceptance letter]

29 Sep 2023

PONE-D-23-04634R1 

Assessing swallowing disorders in adults on high-flow nasal cannula in critical and non-critical care settings. A scoping review protocol 

Dear Dr. Gutierrez-Arias:

I'm pleased to inform you that your manuscript has been deemed suitable for publication in PLOS ONE. Congratulations! Your manuscript is now with our production department. 

Kind regards, 

on behalf of

Dr. Silvia Fiorelli 

Academic Editor

PLOS ONE